# Prevalence of hypertension among adolescents (10-19 years) in India: A systematic review and meta-analysis of cross-sectional studies

Roy Arokiam Daniel[^o], Partha Haldar[iD][^o]*, Manya Prasad[^o], Shashi Kant[^o], Anand Krishnan[^o], Sanjeev Kumar Gupta[^o], Rakesh Kumar[^o]

Centre for Community Medicine, All India Institute of Medical Sciences (AIIMS), New Delhi, India

[^o] These authors contributed equally to this work.
* parthahaldar@outlook.com

## Abstract

### Background

Despite the well-known short-term and long-term ill effects of elevated blood pressure in children and adolescents, pooled data on its burden among Indian adolescents have not yet been synthesized.

### Objectives

We did a systematic review with meta-analysis to calculate the pooled prevalence of hypertension among adolescents (10–19 years) in India.

### Methods

We searched PubMed, Embase, Cochrane library, Google Scholar and IndMed, and included cross-sectional studies reporting data on hypertension prevalence among 10 to19 years old and published in English language from their inception till 1st March 2020. Modified New castle Ottawa scale was used to assess risk of bias based on research design, recruitment strategy, response rate and reliability of outcome determination. A random effects model was used to estimate pooled prevalence, and heterogeneity was assessed using Cochrane's Q statistic test of heterogeneity and $I^2$ statistic. To explore the heterogeneity, we did a meta-regression, and sub-group analyses based on region, study setting and number of blood pressure readings.

### Results

Out of 25 studies (pooled sample of 27,682 participants) six studies were of high, eighteen of moderate, and one was of low quality. The prevalence of hypertension across studies ranged from 2% to 20.5%, with a pooled estimate of 7.6% (95% CI: 6.1 to 9.1%), $I^2$ = 96.6% (p-value <0.001). Sub-group analysis restricted only to the western India demonstrated a

**Data Availability Statement:** All relevant data are within the manuscript and its Supporting Information files.

**Funding:** The author(s) received no specific funding for this work.

**Competing interests:** The authors have declared that no competing interests exist.

**Abbreviations:** BP, Blood pressure; CI, Confidence interval; NCD, Non communicable disease; NHBPEP, National High Blood Pressure Education Program.

smaller heterogeneity ($I^2$ = 18.3%). In univariate model of meta-regression, diagnostic criteria was significantly associated with pooled prevalence (-4.33, 95%CI: -7.532, -1.134)

## Conclusion

The pooled prevalence of hypertension among adolescent in India is 7.6% with substantial heterogeneity between the studies. To tackle the high prevalence of hypertension among adolescents, early detection by screening under school health programme and opportunistic screening at Paediatric OPD should be implemented by Policy makers.

## Introduction

Non communicable diseases (NCDs) accounted for 72% of global deaths in 2016 [1]. In developed nations cardiovascular Diseases (CVD) are one of the major causes of death [2]. Raised blood pressure is a leading risk factor for NCDs [3], which is responsible for 9.2% (95% CI: 8.3 to 10.2%) of DALYs for men and 7.8% (95%CI: 6.9 to 8.7%) of DALYs globally for women in 2015 [4]. Globally, it affects about 1 billion adults and is associated with more than 9 million deaths annually [5]. The prevalence of hypertension among adults is estimated to be 31.1% globally [6] and 27.6% in India [7]. With increasing prevalence, hypertension is becoming a rising health problem not only in adults, but also in children and adolescents [8, 9]. Meta-analysis on hypertension for children and adolescents in Africa showed a pooled prevalence of 5.5% [4, 10].

Cardiovascular disease events are seen most frequently after the fifth decade. Also likely, hypertension in young did not receive a public health attention, which might be due to the lack of awareness [11], considering it as a problem of adults only [12]. However, pathophysiological and epidemiological evidence suggests that essential hypertension and the precursors of cardiovascular diseases such as left ventricular hypertrophy, atherosclerosis and reduced cognitive function [13] originate in childhood but go undetected unless specifically looked for during this age-group [14]. There is strong evidence that raised BMI during adolescence is associated with raised risk of developing hypertension and/or CVD as an adult [15] and also there is a 12% increase in risk of developing CVD for each unit increase in BMI among adolescents [16].

Childhood Blood pressure (BP) is a strong indicator of adult blood pressure, hence, early intervention is important [17]. Thus, early detection of hypertension and its precipitating or aggravating factors is important so that future burden and complications of hypertension can be prevented. In India, the prevalence of hypertension among adolescents, who comprise one-fifth (21%) [18] of India's population, ranges from as low as 2% [19] to 21.5% [20]. Previously, studies have reported pooled country-wide estimate on the prevalence of hypertension among adults [7] and tribal population in India [21], the same among adolescents is lacking. Hence, to overcome this gap, we aimed to perform a systematic review with a meta-analysis of cross-sectional studies to calculate the pooled prevalence of hypertension among adolescents.

## Methods

### Literature search strategy

A comprehensive literature search was carried out between November 2018 to March 2020. The studies published between their inception to 1st March 2020 were searched in Medline via

PubMed, Embase, IndMed, Cochrane library and Google Scholar. The combinations of Medical Subject Headings (MeSH) and free text words (e.g., BP, raised BP, elevated BP, essential hypertension, primary hypertension, and high BP) were combined with search terms related to the outcomes (prevalence, epidemiology, risk). The details about the search strategy is provided in the supplement (S1 Table). We used the PRISMA (Preferred Reporting Items for Systematic reviews and Meta-Analyses) statement for reporting systematic reviews and meta-analyses as a guide for this study [22]. The protocol for the review was registered on the International Prospective Register of Systematic Reviews (PROSPERO) database under the number CRD42019132159.

## Selection criteria

The eligible studies were identified by performing an initial screening of identified titles and abstracts, followed by a full-text review. We included only observational studies adhering to the following criteria: 1) the study was cross-sectional, 2) conducted among adolescent population (10 to 19 years) on prevalence of hypertension, 3) it should be population/community-based (including school-based) studies, 4) sufficient data was available in the article to extract the numerator and denominator for the prevalence of hypertension between 10–19 years and 5) Studies must be in English language. Exclusion criteria were as follows: 1) if studies were conducted exclusively in the age group of less than 10 years or more than 19 years, 2) studies assessing adolescents with specific conditions like adolescents with hypertension or parents with hypertension, obesity, diabetes, chronic kidney disease because studies that included only children pre-disposed to develop secondary hypertension will yield a higher than expected prevalence of hypertension and thus, selection bias and 3) letters, abstracts, conference proceedings, reviews and studies not conducted on humans.

## Study selection

Two independent reviewers (RAD and MP) screened all the titles of retrieved records from the databases, followed by screening of abstracts of relevant titles. Abstracts were selected if they satisfied the selection criteria. Any disagreements about selection were discussed with PH for resolution. All duplicates were removed after verifying the most recent and complete version. Full-text studies were retrieved for the selected abstracts. Reference lists of the retrieved studies were searched (additional sources). The retrieved full text studies were assessed further to ensure they satisfied the inclusion criteria.

## Data extraction

We designed a data collection form in Microsoft Excel [23] to extract and enter the relevant data-fields from the selected full text studies. The data collection sheet included author information, year of publication, study-setting (rural or urban), sampling strategy, sample size, methodology adopted to record blood pressure and the reported prevalence of hypertension. New Castle Ottawa Scale (NOS) [24], modified for cross-sectional study was used to assess the quality of studies included in this review [25]. Studies with score $\geq 8$ were considered high quality, score between 4–7 were considered as moderate quality, and score $\leq 3$ were considered as low-quality studies.

## Statistical analysis

The outcome measure was the prevalence of hypertension. The standard error (SE) of the prevalence was calculated from the reported prevalence, and the sample size for each of the

study, using the formula "square root of p x (1-p)/n. We used 95% confidence interval (CI) to gauge the precision of the summary estimates. The meta-analysis was performed by package *metan* [26] in Stata [27] using random effects model, weighted by inverse of variance. Cochrane's Q statistic test of heterogeneity and $I^2$ statistic (percentage of residual variation attributed to heterogeneity) were performed to evaluate heterogeneity. We reported the pooled prevalence and its 95% confidence intervals (CIs) in the pooled analysis. Publication bias was assessed by visual inspection of funnel plot and small-study effect was assessed by Egger's test. Subgroup analysis was done by zonal divisions of India (region) [28], number of readings and study setting. Sensitivity analysis was done based on the study quality, diagnostic criteria and number of BP readings. Test of interaction was also done to find out if any significant difference was present in the prevalence of hypertension between subgroups. Meta-regression analysis was carried out using *metareg* [29] package of Stata and to explore the cause of heterogeneity using the test by Knapp and Hartung to test the following variables: sample size, mean age, proportion of females, year of publication, diagnostic criteria and number of BP measurements. All the covariates with p-value <0.2 in bivariate model were added to the multivariable model and a p-value <0.05 was considered statistically significant. The goodness of fit of the model was assessed using $R^2$ value and Monte Carlo permutation test was conducted to control for false-positive findings (type I error) when performing meta-regression with multiple covariates. All analyses were performed using the Stata Software (version 13.0).

## Results

### Study selection

Overall, 1707 studies were initially retrieved from the databases. After removing the duplicates, 842 studies were screened and of which, a total of 88 eligible abstracts were screened by inclusion criteria, followed by screening of full text studies. Finally, 25 studies satisfied the eligibility criteria and were included in the meta-analysis (Fig 1).

**Characteristics of studies included in the meta-analysis.** We included a total of 27,682 individuals (females– 41%) in the meta-analysis (Table 1).

Majority of the studies used a simple random sampling strategy to select the study participants, used mercury sphygmomanometer to measure the blood pressure, performed multiple BP recordings, and used NHBPEP criteria to determine the prevalence of hypertension. Kumar [47] did not mention details about the criteria used to classify hypertension; Reddy and Vamsheedar [49] used self-determined cut-off; Gupta et al. [53] used modified ATP classification for classifying hypertension. Majority of the studies (22 of 25 studies) were school based; Savitha [33] and Mujumdar [19] were community-based. Except the studies by Rai [52] and Kumar [50], remaining studies were done in urban setting. All the studies measured blood pressure using mercury sphygmomanometer except the study done by George [44] and Kumar [47] who had used digital blood pressure apparatus. All the studies collected data prospectively. None of the studies were multisite or nationally representative.

### Risk of bias assessment

Out of 25 studies, six studies were of high quality, eighteen were of moderate quality and one study was low quality (S2 Table). Among the included studies, 14 studies had used validated tool, acceptable non-response rate and robust sampling strategy.

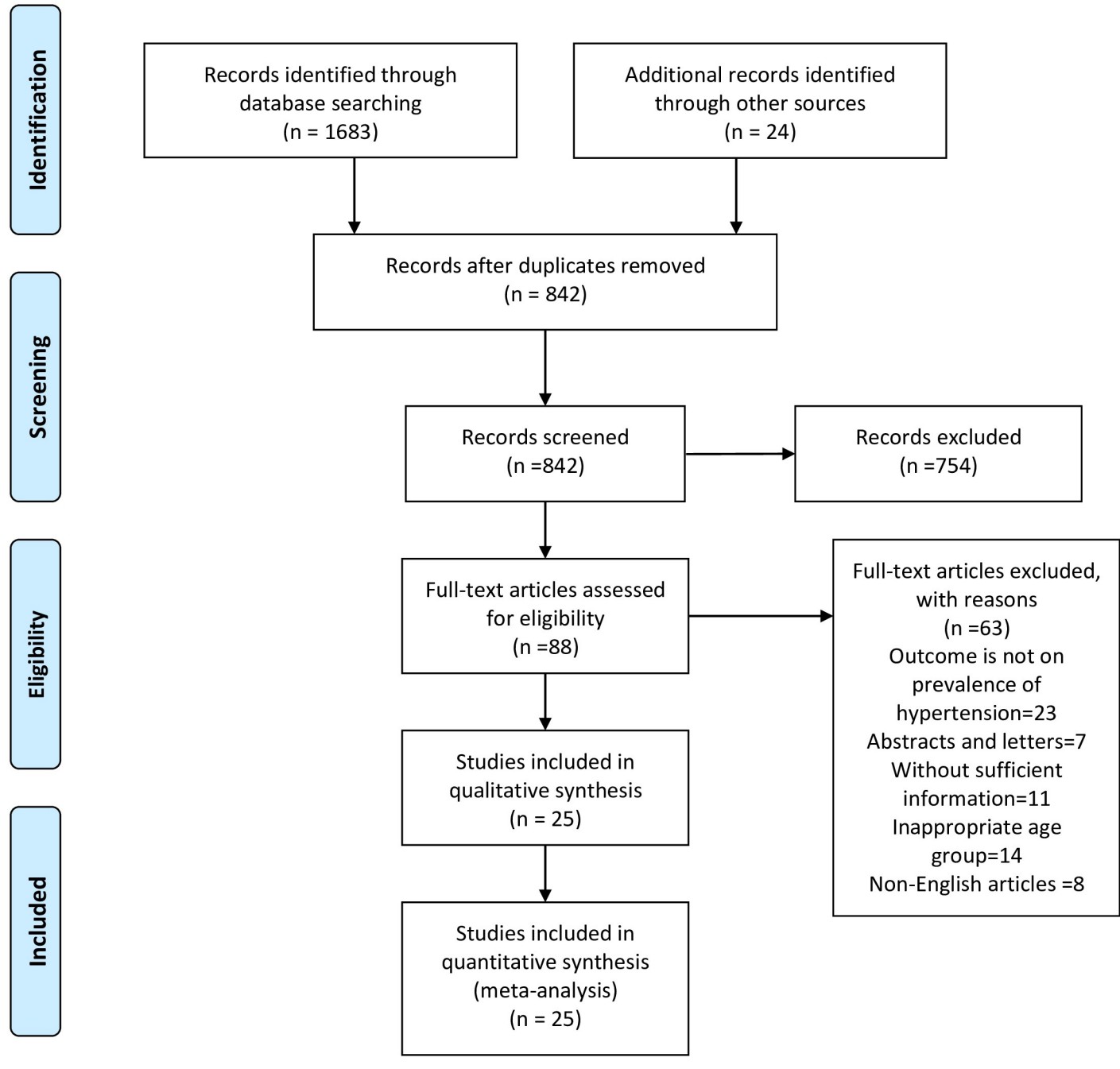

**Fig 1. Flow of selection of studies for meta-analysis.**

## Prevalence of hypertension among adolescents in India

Prevalence of hypertension for both sexes combined (n = 25 studies) ranged from 2% in a cross-sectional study done by Mujumdar [19] in Southern India, to 20.5% by Gupta et al. [53] conducted in Northern India.

**Random effects pooled estimate.** The random effects pooled estimate for prevalence of hypertension among adolescent in India was 7.6% (95% CI: 6.1 to 9.1%) (Fig 2). There was

**Table 1. The characteristics of the studies included in the systematic review and meta-analysis.**

| S. No | Author & year of publication | Study area | Study setting | Sample size | Age group included (in years) | No. of mean readings | Sampling strategy | Diagnostic criteria for hypertension | Prevalence of hypertension (%) |
|---|---|---|---|---|---|---|---|---|---|
| 1. | Mohan et al. 2004 [30] | Punjab | Urban and rural, school-based | 3326 | 11–17 | 2 | NA* | #NHBPEP | 5.7 |
| 2. | Anjana 2005 [31] | Punjab | Urban, school-based | 529 | 6–14 | 3 | NA | NHBPEP | 8.1 |
| 3. | Saha 2007 [32] | Kolkata | Urban, community-based | 1081 | 10–19 | 3 | Simple random sampling | NHBPEP | 2.9 |
| 4. | Savitha 2007 [33] | Karnataka | Urban, school-based | 503 | 10–16 | 3 | Stratified random sampling | NHBPEP | 6.2 |
| 5. | Sharma 2009 [34] | Himachal Pradesh | Urban and rural, school-based | 1085 | 11–17 | 3 | NA | NHBPEP | 5.9 |
| 6. | Goel 2010 [35] | Delhi | Urban, school-based | 1022 | 14–19 | 2 | Multistage cluster sampling | NHBPEP | 6.4 |
| 7. | Khan 2010 [36] | Gujarat | Urban, school-based | 1093 | 12–19 | 2 | Simple random sampling | NHBPEP | 9.8 |
| 8. | Buch 2011 [37] | Gujarat | Urban, school-based | 535 | 6–18 | 3 | Purposive sampling | NHBPEP | 8.8 |
| 9. | Durrani and Waseem 2011 [38] | Uttar Pradesh | Urban, school-based | 701 | 12–16 | 3 | Stratified random sampling | NHBPEP | 9.4 |
| 10. | Mujumdar 2012 [19] | Karnataka | Urban, school-based | 772 | 6–15 | NA | NA | NHBPEP | 2.0 |
| 11. | Kumar 2012 [39] | Maharashtra | Rural, community-based | 1055 | 10–19 | 3 | Simple random sampling | NHBPEP | 3.4 |
| 12. | Yuvaraj 2014 [40] | Karnataka | NA, school-based | 1732 | 9–16 | 3 | NA | NHBPEP | 2.5 |
| 13. | Lone 2014 [41] | Maharashtra | Urban, school-based | 540 | 12–16 | 3 | Simple random sampling | NHBPEP | 11.8 |
| 14. | Anand 2014 [42] | New Delhi | Urban, school-based | 315 | 12–17 | 2 | Complete enumeration | NHBPEP | 7.0 |
| 15. | Faujdar 2014 [43] | Maharashtra | Urban, school-based | 999 | 11–17 | One | Complete enumeration | NHBPEP | 11.2 |
| 16. | George 2014 [44] | New Delhi | Urban, school-based | 485 | 9–18 | NA | Convenience sampling | NHBPEP | 8.2 |
| 17. | Garg 2015 [45] | Uttar Pradesh | Urban, school-based | 1000 | 10–14 | 3 | NA | NHBPEP | 9.4 |
| 18. | Mahajan and Negi 2015 [46] | Himachal Pradesh | Urban, school-based | 3385 | 10–19 | 3 | Simple random sampling | NHBPEP | 11.3 |
| 19. | Kumar 2015 [47] | Puducherry | Urban and rural, school-based | 1100 | 11–17 | 3 | Stratified random sampling | Not Available | 4.1 |
| 20. | Maiti and Bandyopadhyay 2016 [48] | West Bengal | Urban, school-based | 129 | 10–19 | Last 2 | Simple random sampling | NHBPEP | 10.1 |
| 21. | Reddy and Vamsheedar 2017 [49] | Andhra Pradesh | Urban, school-based | 568 | 5–14 | NA | NA | Self-determined cut-off | 4.4 |
| 22. | Kumar 2017 [50] | Bihar | Rural, school-based | 2913 | 13–15 | 3 | Stratified cluster sampling | NHBPEP | 4.6 |
| 23. | Singh et al. 2017 [51] | Madhya Pradesh | Urban, school-based | 404 | 10–18 | 3 | Simple random sampling | NHBPEP | 15.3 |

*(Continued)*

**Table 1.** (Continued)

| S. No | Author & year of publication | Study area | Study setting | Sample size | Age group included (in years) | No. of mean readings | Sampling strategy | Diagnostic criteria for hypertension | Prevalence of hypertension (%) |
|---|---|---|---|---|---|---|---|---|---|
| 24. | Rai 2018 [52] | Karnataka | Rural, school-based | 400 | 8–17 | 3 | Stratified random sampling | NHBPEP | 4.3 |
| 25. | Gupta et al. 2018 [53] | Himachal Pradesh | Urban, school-based | 2100 | 10–16 | 2 | Population proportionate to size | Modified ATP@ classification | 20.5 |

[#]NHBPEP- National High Blood Pressure Education Program.

[@]ATP- Adult Treatment Panel.

[*]NA- Not available.

significant heterogeneity between the studies. Heterogeneity test showed $I^2$ = 96.7%, Q = 723.8 and p-value <0.001.

## Subgroup analysis

**Prevalence of hypertension based on geographical region.** Based on the zonal divisions of India, places of study were grouped into five regions: north, south, east, central, and west (Fig 3). Studies conducted in west region demonstrated a small heterogeneity ($I^2$ = 18.3%, p-value = 0.294). There was significant difference in the prevalence in the studies conducted among the various regions in India (p-value<0.001). The prevalence of hypertension among various sub-groups is shown in Table 2.

**Prevalence of hypertension based on number of blood pressure readings.** Out of 25 studies, three studies did not mention about the number of readings and one study with only one reading was excluded from the analysis. There was significant difference in the prevalence between the studies that used two and three readings to classify hypertension as shown in Fig 4 (p-value<0.001).

**Prevalence of hypertension based on study setting.** Out of 25 studies, three studies were conducted in both urban and rural regions of which one study did not mention details on sample size and prevalence of hypertension based on the setting. There was significant difference in the prevalence between the studies of rural and urban setting as shown in Fig 5 (p-value<0.001).

## Publication bias

Funnel plot demonstrated a mild asymmetry (Fig 6). However, the p-value for Egger's test was observed to be 0.288, implicating no or undetected publication bias.

## Sensitivity analysis

Sensitivity analysis was performed by removing one low quality study by Mohan et al. and the prevalence of hypertension showed no substantial change from 7.6% (95% CI: 6.1 to 9.1%) to 7.7% (95% CI: 6.1 to 9.4%). The pooled estimate after removal of three studies that used criteria other than NHBPEP criteria to diagnose hypertension was 7.3% (95%CI: 5.9 to 8.6%) The pooled estimate after removing 10 studies based on number of blood pressure readings, (3- no. of readings not mentioned, 1- with single reading and 6- with only two readings) was 7.0% (95% CI: 5.3 to 8.7%) as shown in Table 3.

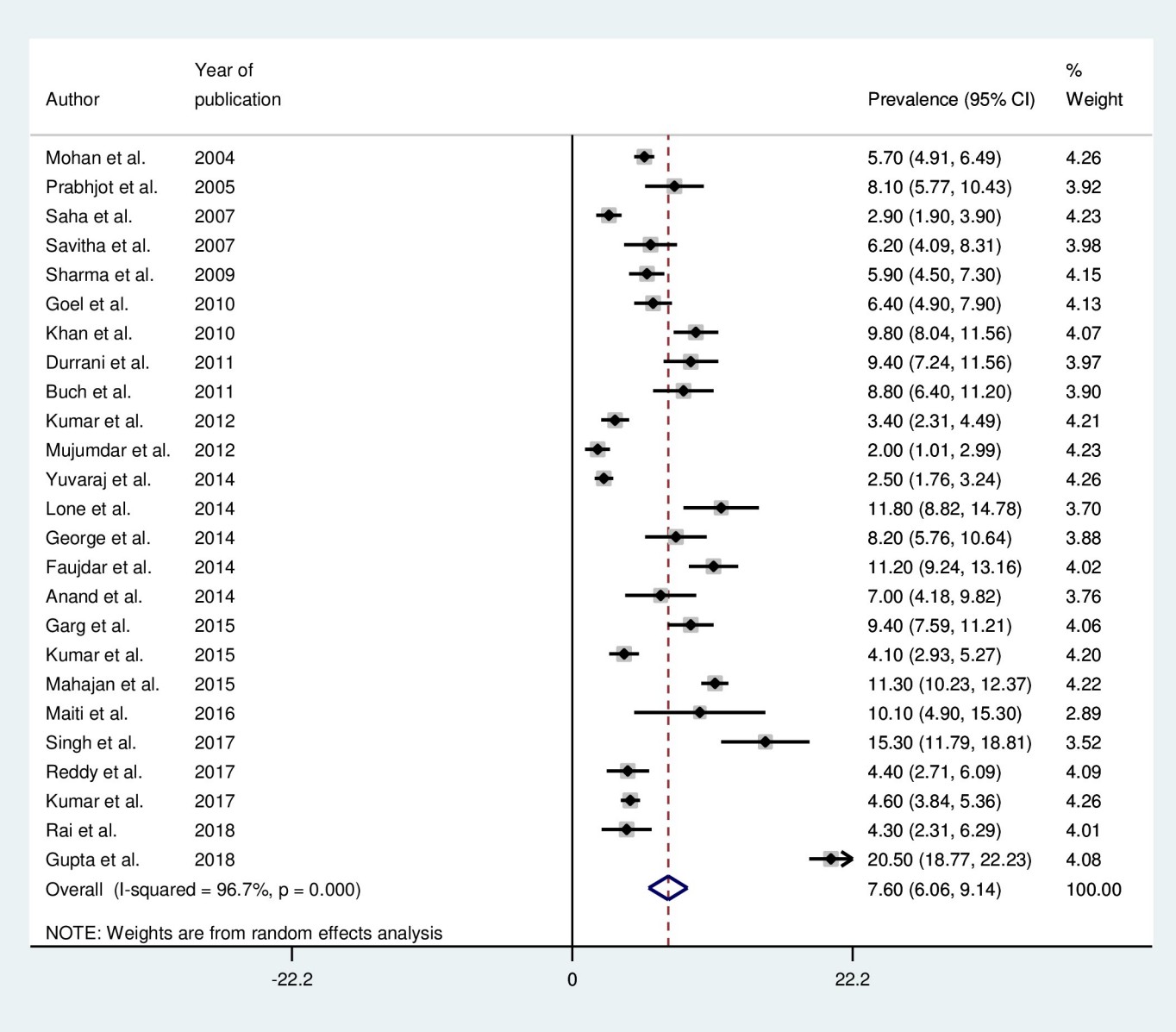

**Fig 2. Forest plot of pooled prevalence of hypertension among adolescents in India.**

## Meta regression

In the univariate meta-regression, we observed that change in diagnostic criteria caused a decrease in the effect size with the beta-coefficient value -4.33(-7.532, -1.134). The multivariate model with year of study, number of BP readings, and diagnostic criteria were included in which none of the covariates came out to be statistically significant (Table 4). The $R^2$ value for meta-regression was 23%. Monte Carlo permutation test was conducted with 50,000 permutations and the adjustment for type 1 error did not change the result.

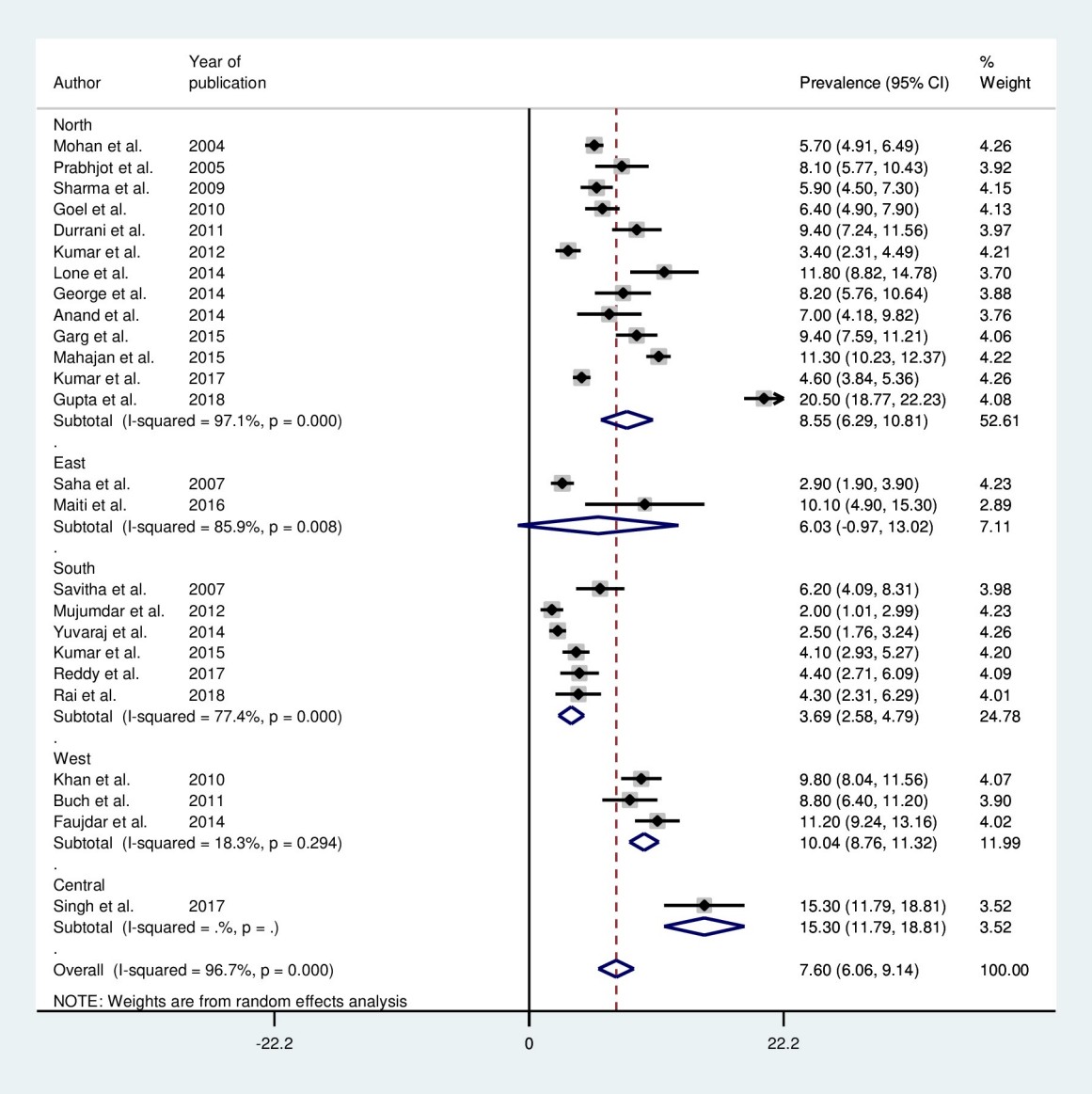

**Fig 3. Forest plot of region wise prevalence of hypertension among adolescents in India.**

## Discussion

We did a systematic review and meta-analysis of data from 25 studies involving 27,682 participants and found a pooled prevalence of hypertension of 7.6% (95% CI: 6.1 to 9.1%) among adolescents in India. The studies had significant statistical heterogeneity between them, which we could partly explain by subgroup analysis. The studies included from Western India demonstrated quite small heterogeneity.

A systematic review and meta-analysis done by de Moraes [54], reported prevalence of hypertension of 11.2% among adolescents from developed and developing countries which was higher that the pooled prevalence in our study (7.6%). The possible reason for disparity in the estimate of hypertension prevalence might be due to the inclusion of hypertension

**Table 2. Prevalence of hypertension among adolescents by sub-groups.**

| Sub-group | No. of studies | Total no. of participants | Prevalence (%) | 95% CI | Heterogeneity test | | p-value* (subgroup difference) |
|---|---|---|---|---|---|---|---|
| | | | | | $I^2$ | Q | |
| **Region** | | | | | | | |
| Central | 1 | 404 | 15.3 | 11.8–18.8 | - | - | <0.001 |
| West | 3 | 2,627 | 10.0 | 8.8–11.3 | 18.3 | 2.5 | |
| North | 13 | 18,366 | 8.6 | 6.3–10.8 | 97.1 | 413.7 | |
| East | 2 | 1,210 | 6.0 | -0.9–13.0 | 85.9 | 7.1 | |
| South | 6 | 5,075 | 3.7 | 2.6–4.8 | 77.4 | 22.1 | |
| **No. of readings** | | | | | | | |
| Two readings | 6 | 7,985 | 9.9 | 5.1–14.7 | 97.9 | 243.4 | <0.001 |
| Three readings | 15 | 16,873 | 7.0 | 5.3–8.7 | 95.6 | 317.0 | |
| **Study setting** | | | | | | | |
| Rural | 5 | 5,794 | 3.8 | 2.9–4.7 | 60.6 | 10.2 | <0.001 |
| Urban | 20 | 19,105 | 8.8 | 6.7–10.8 | 96.5 | 542.7 | |

*Null hypothesis for this test is that there is no difference in the subgroups and the test of significance carried was out by chi square test.

estimates from regions like North America, Oceania, Africa, Europe and Latin America in their meta-analysis. They also included studies using a cut-off of more than 90[th] percentile for elevated blood pressure which might have overestimated the prevalence of elevated blood pressure in their study [55] whereas, the studies included in this systematic review and meta-analysis have used a cut-off of more than 95[th] percentile. The difference might also be attributed to the race of the population studied. The former study mainly consisted of Mongoloid race, whereas the present study included population belonging to Mongoloid, Dravidian and Caucasoid [56, 57]. A systematic review and meta-analysis conducted by Noubiap JJ et al. [4] among African children and adolescents of age group 2–19 years, estimated the prevalence of hypertension as 5.5% (95% CI: 4.2 to 6.9%) which is lower from the current estimate of 7.6%, which might be due to the age group considered in the previous study. Genetic factors have been documented to play a major role in determining hypertension prevalence. One such factor is the Gly460Trp allele, which is implicated in being responsible for a higher prevalence of hypertension in Asians [58–60].

Another systematic review and meta-analysis of hypertension prevalence done among the Brazilian adolescents (10–19 years) by Goncalves [61] reported a pooled estimate of prevalence of 8.0% (95% CI 5.0 to 11.0%). This estimate is similar to our study finding which might be due to Asian contribution to the Brazilian population [62].

Existing information on prevalence of hypertension has methodological issues like different criteria used for diagnosis of hypertension, number of readings and instrument used to record blood pressure. Majority of the studies did not mention any details about the calibration of the blood pressure apparatus. According to the criteria of the European Hypertension Society, and the American Academy of Pediatrics recommended that difference between averages of the measure's mercury column and tested monitor for a device to be validated should be ≤ 5 mmHg [63, 64]. Also, that the standard deviation of the differences of the averages should not be larger than 8 mmHg. The differences in the prevalence can introduce misclassification of individuals and may cause underestimation or overestimation of the true prevalence [65].

Majority of the studies were school based and the pooled prevalence of hypertension among school-based studies was higher than the community-based studies. Some factors may

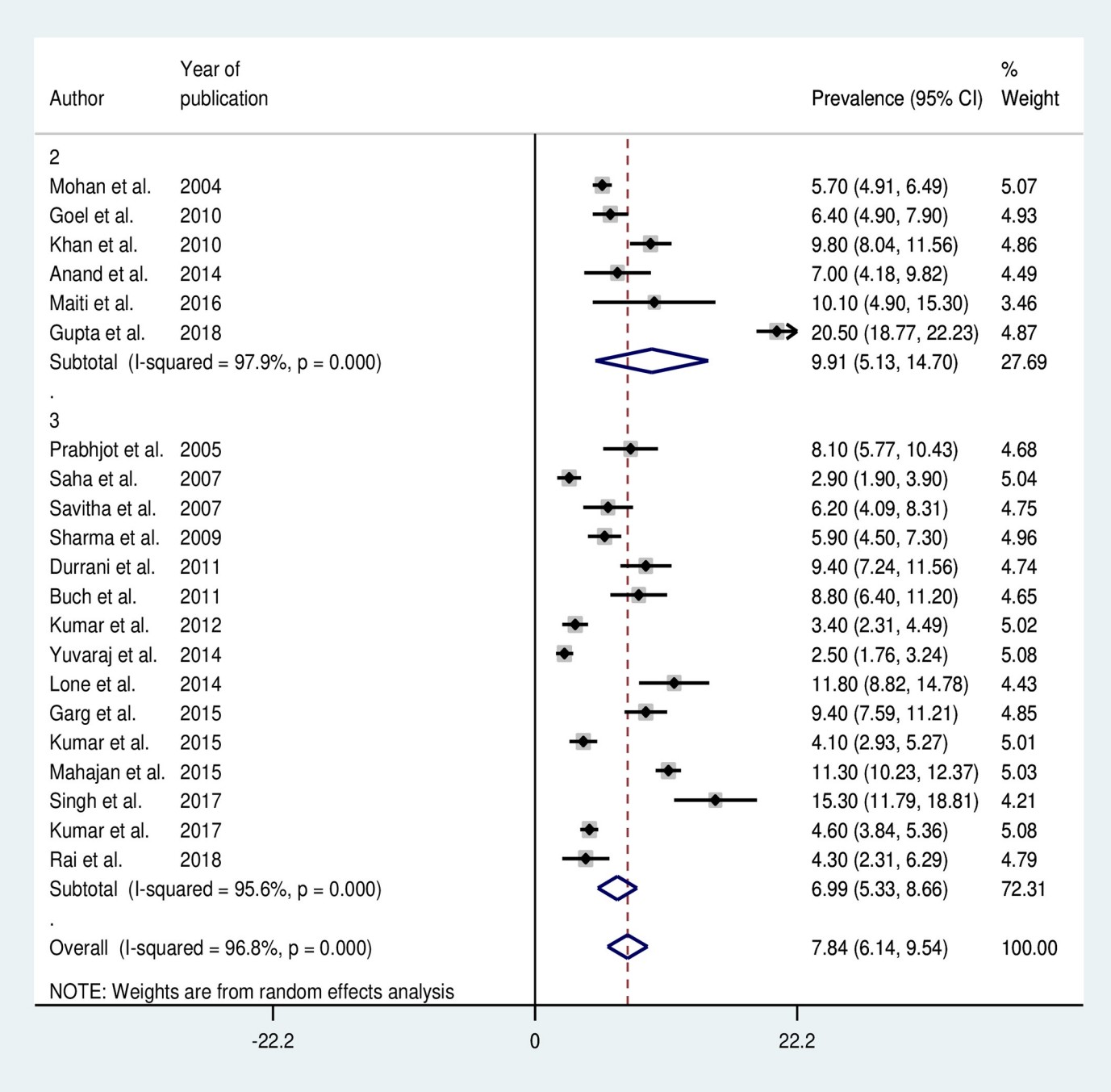

**Fig 4. Forest plot of prevalence of hypertension with two and three blood pressure readings among adolescents in India.**

influence the blood pressure levels when it is measured in school setting. Talking or active listening is considered as one major factor that increases the level of blood pressure [66]. Full bladder tends to increase the blood pressure by 10-15mmHg [67]. The stress and anxiety (due to tests and assessments in school) in students could be responsible for the increased blood pressure [68]. Also, the school enrolment ratio is less especially in the rural parts of India and a

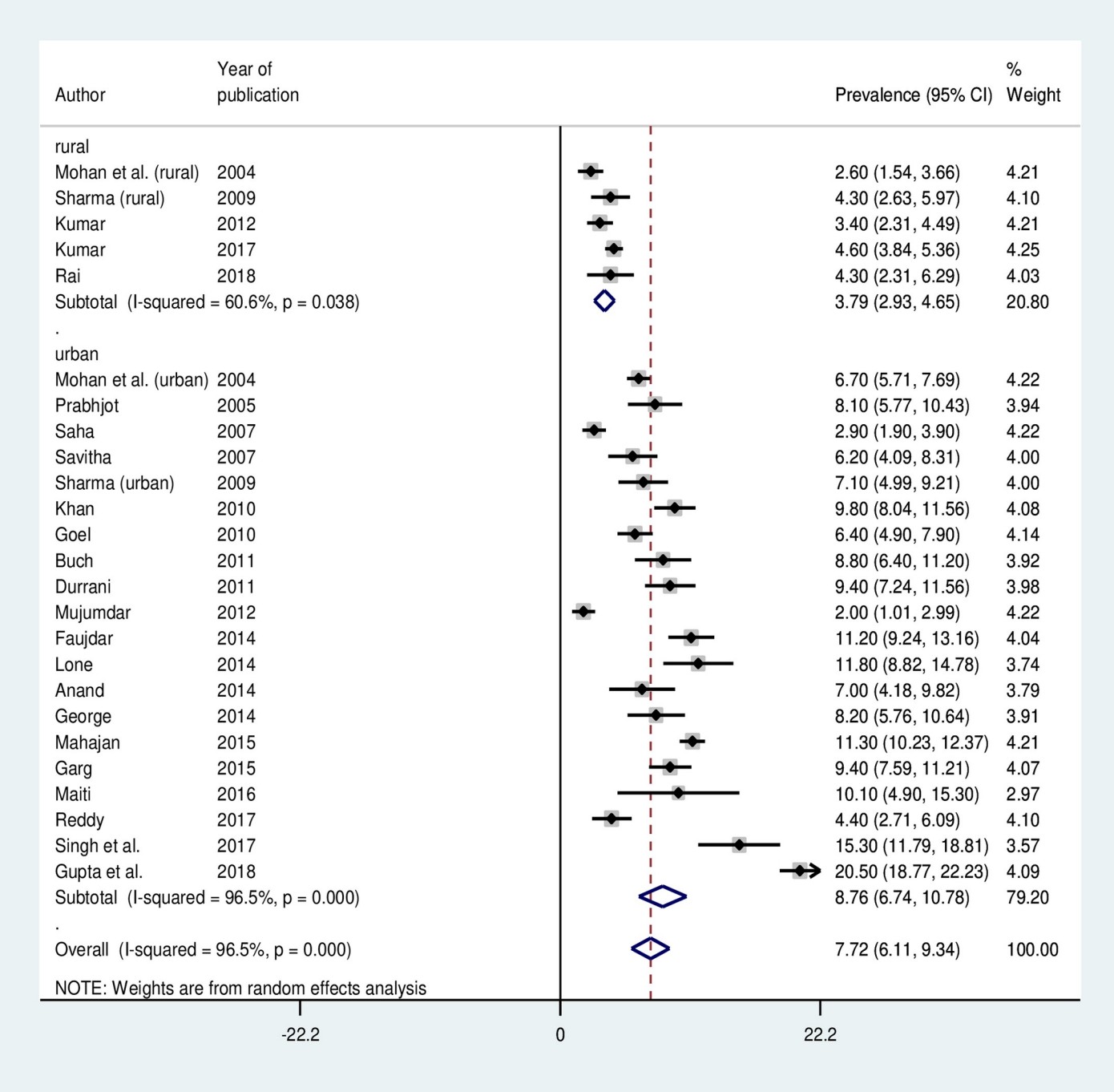

**Fig 5. Forest plot of prevalence of hypertension among adolescents in India based on study setting.**

school-based study will not be representative of the population [69]. Hence with all these factors community-based studies appear to be superior than school-based studies.

Most of the studies were from the urban area and only three studies were from the rural areas. Urban studies revealed higher prevalence of hypertension compared to the studies from rural parts of India which might be due to the sedentary lifestyle of urban participants. Higher

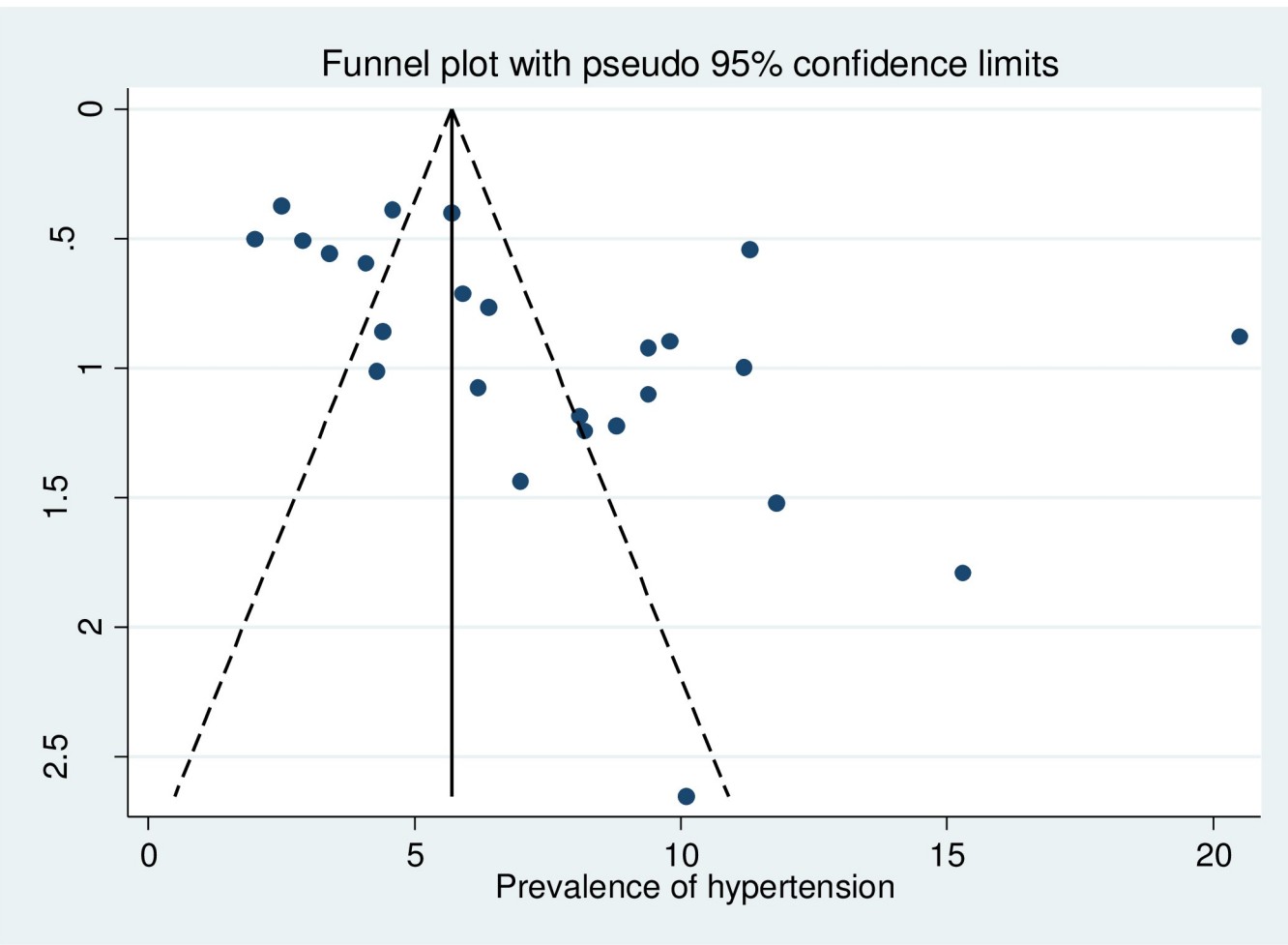

**Fig 6. Funnel plot to check for publication bias.**

prevalence of hypertension was seen in studies conducted in North and West part of India than in the studies conducted in South India.

High heterogeneity was present in the pooled as well subgroup analyses, which requires caution in extrapolating the results. In sub-group analysis, west region showed a small heterogeneity ($I^2$ = 18%). Other sub-groups on various variables showed high heterogeneity. The reasons for high heterogeneity could be different sampling strategy, methodology of blood pressure measurement, regional variations and varied cut-off used to diagnose hypertension (e.g. NHBPEP criteria, modified ATP criteria and cut-off arbitrarily decided by authors [49, 52, 53]). Other reasons could be natural differences among the individuals included in the

**Table 3. Results for sensitivity analysis for the prevalence of hypertension among adolescents.**

| S. No | Sensitivity analysis | Prevalence of hypertension (%) with 95% CI | Heterogeneity test | | p-value |
|---|---|---|---|---|---|
| | | | $I^2$ | Q | |
| 1. | Removing one low quality study | 7.7 (6.1–9.4) | 96.7 | 723.8 | <0.001 |
| 2. | Removing studies that have used criteria, other than NHBPEP, to diagnose hypertension | 7.3 (5.9–8.6) | 95.1 | 428.6 | <0.001 |
| 3. | Removing studies that included only one and two readings of blood pressure | 7.0 (5.3–8.7) | 95.6 | 317.0 | <0.001 |

**Table 4. Results for meta-regression for the prevalence of hypertension among adolescents.**

| S.No | Covariate (No. of studies included) | Univariate model | | Multivariate model | |
|---|---|---|---|---|---|
| | | Meta-regression coefficient (95%CI) | p-value | Meta-regression coefficient (95%CI) | p-value |
| 1. | Year of publication (25) | 0.321(-0.139, 0.692) | 0.148 | 0.239(-0.203, 0.681) | 0.271 |
| 2. | Sample size (25) | 0.000(-0.001, 0.002) | 0.833 | - | |
| 3. | Mean age (11) | -0.511 (-3.184, 2.163) | 0.676 | - | |
| 4. | No. of BP reading (22) | -2.478 (-5.770, 0.813) | 0.132 | -2.053 (-5.064, 0.958) | 0.169 |
| 5. | Diagnostic criteria (24) | -4.33 (-7.532, -1.134) | 0.010 | -3.423 (-7.354, 0.508) | 0.084 |
| 6. | Proportion of females (23) | 0.000 (-0.372, 0.373) | 0.984 | - | |

studies since states and cities are socioeconomically and culturally different from each other. In meta-regression, none of covariate was statistically significant.

One of the potential limitations was the blood pressure measurement methods used in the studies included in the review. Measurements varied significantly among the various studies and in relation to their adaptations and interpretations, which may influence the summarization of the prevalence. The NHBPEP criteria recommends that at least three blood pressure readings should be recorded and for diagnosis, this should be repeated for at three different occasions [70]. But majority of the studies had not adhered to this recommendation.

NHBPEP includes overweight children in the blood pressure distribution data and uses data of the first blood pressure reading only. Some studies have found that inclusion of overweight/obese children raised the cutoff points for elevated blood pressure [71–75]. Xi et al. have now established an international blood pressure reference, based on data from seven large cross-sectional surveys consisting of non-overweight children and adolescents. When compared with the US fourth report at median height, systolic BP of the corresponding percentiles of these international references was lower, whereas diastolic BP was similar [76].

The various factors that are related to blood pressure measurement were cuff size (small cuff-size overestimates blood pressure [77, 78]), technique used to determine the DBP (choice of fourth or fifth Korotkoff sound), the number of BP measurements, and type of instruments used (oscillometric or mercury sphygmomanometer). In a systematic review and meta-analysis conducted among children showed that higher SBP readings were recorded by oscillometric devices as compared to a standard mercury sphygmomanometer with a pooled effect estimate of 2.53 mmHg, 95% CI 0.57 to 4.5 mmHg) [79]. Another comparative study conducted by Shahbabu to estimate the accuracy of readings of aneroid and digital sphygmomanometers in reference to mercury sphygmomanometers showed that more than 89% of aneroid readings and less than 44% of the readings by digital device had absolute difference of 5mm Hg when compared with the mercury readings for both systolic and diastolic blood pressure. Sensitivity and specificity of aneroid device was higher (86.7% and 98.7%) than digital device (80% and 67.7%) [80]. In a multicentric study conducted among children with chronic kidney disease (CKD), oscillometric SBP and DBP measurements were constantly higher than the readings obtained by auscultation (median elevations of 9 and 6 mm for SBP and DBP) [81]. So, this overestimation of both systolic and diastolic blood pressure leads to frequent misclassification of blood pressure values with false-positive diagnosis of hypertension. Moreover, oscillometric devices require regular maintenance and repeated calibration for accurate BP measurements. Hence these factors should be given due attention for the accurate diagnosis of hypertension which plays a pivotal role in the blood pressure measurement and to prevent overdiagnosis.

A study done by Raj M et al., published in 2010, a cross sectional study in Kerala among 5–16 years of age, to determine blood pressure distribution in schoolchildren and to derive

population specific reference values appropriate for age, gender, and height status. They found that these children exhibited higher diastolic blood pressures for both boys and girls than the US children across all age groups and for systolic blood pressure, girls showed higher values than the international standard while for boys, there was a minimal difference [82]. But this study has considered only for school going children and was not a community-based study which might not be representative of all the children of that age group.

Cost-effectiveness study conducted in United States published in 2011, about the blood pressure screening in adolescents showed that the population-wide strategies such as salt reduction (cost-saving [boys] and $650/ QALY [girls]) and increasing physical education ($11 000/QALY [boys] and $35 000/QALY [girls]) and treating the adolescents at highest risk was most cost-effective [83]. Hence population-based high-risk screening should be undertaken which would be an effective solution to prevent hypertension and future burden of cardiovascular diseases.

Also, studies show that oscillometeric devices overestimates blood pressure when compared with mercury sphygmomanometer [84]. NHBPEP reference blood pressure tables were based on mercury sphygmomanometer so; the prevalence of hypertension increases if NHBPEP criteria was followed when other oscillometeric devices were used to measure the blood pressure. All the children above 3 years of age should have an annual blood pressure examination. The confirmatory diagnosis and appropriate management should be not be made by automatic blood pressure apparatus as per the guidelines [70].

Differences in study setting and the lack of method standardization reflected in different equipment and different intervals between measurements, may have contributed to the observed heterogeneity. The heterogeneity could not be explained by subgroup analyses or meta-regression. It is possible that other subject characteristics such as nutritional status, stages of adolescence and sexual maturity may have a role to play in determining prevalence of hypertension. However, the absence of this information in most original studies prevented further analysis. Signs and symptoms in the early stages of hypertension do not present unless micro or macrovascular complications occur, thus it is also called as 'silent killer' [85]. Hence, health promotion is more relevant today in addressing NCDs. Health educational programmes should be implemented across all the schools and basic awareness on prevention of hypertension should be imparted to the adolescents in school. It will not only educate the adolescent and modify their behaviors; it would also have an impact over the knowledge on hypertension of their parents who might be in the early stages of development of hypertension. So, health promotion for school students through school health programme and through Anganwadi constitutes an important strategy for behavior change communication. School based aerobic exercise programmes that are proven effective in reducing risk may be implemented [86]. A familial tendency for developing high blood pressure is well known, which suggests a genetic role in the development of hypertension [87]. Hence, adolescents having a positive family history of hypertension, should be screened which helps in early diagnosis and appropriate management which in turn helps in reducing the burden and complications of hypertension.

## Strengths and limitations

To the best of our knowledge, our's is a first systematic review and meta-analysis that estimated the prevalence of hypertension among adolescents in India. We used a standard search strategy, risk of bias assessment for individual studies, and explored heterogeneity using subgroup analysis, and meta-regression. But a few limitations were there in our study. We did not consider, studies reported in language other than in English, and grey literature. We believe this not to affect our findings since in India, almost all medical literature is published in

English language. The pooled estimate of hypertension emerging from this study needs to be interpreted along with the considerable heterogeneity observed between the studies. The number of studies conducted in rural settings that were included in this review were less in number and hence it limits the generalizability of the results as it is an important determinant.

## Conclusion

The pooled prevalence of hypertension among adolescent in India is 7.6% with substantial heterogeneity between the studies. Hypertension in adolescents poses an important issue in public health and clinical medicine. Due attention should be given to this growing concern. Future studies should evaluate the use of a screening programme for hypertension in schools and community. Randomized Control Trails (RCTs) evaluating the effect of screening on relevant outcomes like CVD events and mortality would be instrumental in guiding future policy. A uniform criterion to classify hypertension in children and adolescents ought to be developed through large community-based studies in India rather than classifying hypertension based on US reference population. Early detection by screening for hypertension among students under school health programme and opportunistic screening at Paediatric OPD should be implemented by Policy makers.

### Perspectives

There is no country wide information from India on adolescent hypertension. This review provides the pooled estimate on the prevalence of hypertension but also highlights the heterogeneity in the studies conducted across India, which to needs to be interpreted with caution.

## Supporting information

**S1 Checklist. PRISMA 2009 checklist.**
(DOC)

**S1 Table. Search strategy.**
(DOCX)

**S2 Table. Risk of bias assessment for all the selected studies for systematic review and meta-analysis.**
(DOCX)

## Author Contributions

**Conceptualization:** Roy Arokiam Daniel, Partha Haldar.

**Data curation:** Roy Arokiam Daniel, Manya Prasad.

**Formal analysis:** Roy Arokiam Daniel, Manya Prasad.

**Investigation:** Roy Arokiam Daniel, Manya Prasad.

**Methodology:** Roy Arokiam Daniel, Manya Prasad.

**Project administration:** Partha Haldar.

**Resources:** Roy Arokiam Daniel, Manya Prasad.

**Software:** Roy Arokiam Daniel, Manya Prasad.

**Supervision:** Partha Haldar, Shashi Kant, Anand Krishnan, Sanjeev Kumar Gupta, Rakesh Kumar.

**Validation:** Partha Haldar, Shashi Kant, Anand Krishnan, Sanjeev Kumar Gupta, Rakesh Kumar.

**Visualization:** Partha Haldar, Shashi Kant, Anand Krishnan, Sanjeev Kumar Gupta, Rakesh Kumar.

**Writing – original draft:** Roy Arokiam Daniel, Manya Prasad.

**Writing – review & editing:** Roy Arokiam Daniel, Partha Haldar, Manya Prasad, Shashi Kant, Anand Krishnan, Sanjeev Kumar Gupta, Rakesh Kumar.

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
