## [Decision Letter · Decision Letter 0]

10 Mar 2020

PONE-D-20-01922

Prevalence of hypertension among adolescents (10-19 years) in India: A systematic review and meta-analysis of observational studies

PLOS ONE

Dear Dr. Haldar,

Thank you for submitting your manuscript to PLOS ONE. After careful consideration, we feel that it has merit but does not fully meet PLOS ONE’s publication criteria as it currently stands. Therefore, we invite you to submit a revised version of the manuscript that addresses the points raised during the review process.

We would appreciate receiving your revised manuscript by Apr 24 2020 11:59PM. To enhance the reproducibility of your results, we recommend that if applicable you deposit your laboratory protocols in protocols.io, where a protocol can be assigned its own identifier (DOI) such that it can be cited independently in the future. For instructions see: http://journals.plos.org/plosone/s/submission-guidelines#loc-laboratory-protocols

We look forward to receiving your revised manuscript.

Kind regards,

Simeon-Pierre Choukem

Academic Editor

PLOS ONE

Journal Requirements:

2. We noticed that your last search for your systematic review was performed in January 2019. Please ensure that that you search is up to date and the systematic review/meta-analysis includes any studies published since then.

Reviewers' comments:

Reviewer's Responses to Questions

**Comments to the Author**

1. Is the manuscript technically sound, and do the data support the conclusions?

Reviewer #1: Partly

Reviewer #2: Yes

2. Has the statistical analysis been performed appropriately and rigorously? 

Reviewer #1: No

Reviewer #2: Yes

3. Have the authors made all data underlying the findings in their manuscript fully available?

Reviewer #1: Yes

Reviewer #2: Yes

4. Is the manuscript presented in an intelligible fashion and written in standard English?

Reviewer #1: Yes

Reviewer #2: No

5. Review Comments to the Author

Reviewer #1: Title

1) I suggest to the authors to replace in the title the term “observational” by “cross-sectional” since in inclusion criteria, they considered only cross-sectional studies.

Introduction

2) The sutdy is well justify and the objective is clear.

Methods

3) The authors should update the stage of the review in PROSPERO. For instance, “Preliminary searches” are not “completed”.

4) Inclusion. It is not clear whether the authors only considered population/community-based studies (including school-based) studies. Only this kind of studies should be considered to have reliable epidemiological data.

5) Data extraction. Cochrane do not recommend scoring quality assessment (https://handbook-5-1.cochrane.org/chapter_8/8_assessing_risk_of_bias_in_included_studies.htm).

6) Statistical analysis. Consider reporting all prevalence estimates with 95% prediction intervals that better measure uncertainty.

Results

7) Characteristics of included studies. The sentence “All included studies were cross-sectional studies.” is not necessary since in the Inclusion criteria, the authors stated that they will consider only cross-sectional studies.

8) Table 1 should include diagnostic criteria for hypertension.

9) Table 2. The signification of 0, 1, 2, 3, 4 for Selection, Comparability, and Outcome is not clear. This was not described any part in manuscript.

10) Please report all prevalence with only one decimal.

11) Results. Line 285: “a little increase”. I’m not sure it is adequate to write a little increase. I would suggest changing to “no substantial change”.

12) Results. Line 286-289: “Analysis repeated after removing two studies that reported the hypertension prevalence of more than 15% and the prevalence of hypertension decreased from 7.64% (95% CI: 6.13% to 9.14) to 6.75% (95% CI: 5.55 to 7.96%) with more precise estimate.” It is not a correct methodological practice to exclude studies based on the outcome of outcome (here prevalence). Sensitivity analyses are carried out on methodological criteria, criteria for inclusion of participants and methodological quality (risk of bias). The authors may explore why these studies had higher prevalence when looking at participants characteristics, method (example sampling method), setting (urban vs rural for example)… Authors can conduct leave-one-out sensitivity analysis to assess the effect on the overall pooled estimate of removing each study.

13) Results. Line 289-290: “We also reran the main analysis by removing two community-based study and one hospital-based study”. The authors should not pool community-based and hospital-based studies. In addition, in term of epidemiology only community-based studies should be considered to have reliable hypertension prevalence in the adolescent populations. In general, burden of diseases is higher in hospital-based studies and do not represent the “real” face of the epidemiology in the population. In conclusion, hospital-based studies should be removed from this meta-analysis.

14) Results. Line 289-290: “We also reran the main analysis by removing two community-based study and one hospital-based study”. It is not clear why the authors removed both community-based and hospital-based studies.

15) Results. Line 291-293. “The analysis was repeated after removing the study done by Maiti & Bandyopadhyay with sample size of less than 200 and the pooled estimate came out to be 7.57%, 95% CI: 6.04 to 9.09%.” The authors should give a strong rationale on the reason for choosing a threshold of 200.

16) Results. Table 5.

a. It is not whether the findings in this table are for univariable or multivariable meta-regression. Both models (uni- and multivariable) should be reported.

b. I do not understand why some variance are reported with “(-)”. Variance is never negative.

c. P value less than 0.05 should be reported with at least three decimals.

d. Meta-regression should include methods to diagnose hypertension i.e. diagnostic criteria.

e. For total no. of males and females, only the proportion of males or females should be considered since the no. can vary among studies, but the proportion is a comparable measure of the representativeness of sex distribution.

f. How many studies were included in final multivariable model?

g. How many studies were included in each univariable model?

h. It is not clear why for the “Age group” variable, the authors reported only one coefficient since there would be n-1 coefficients according to the number age groups.

Discussion

17) Line 318-320. “The possible reason for disparity in the estimate of hypertension prevalence might be due to the race and ethnicity of the developed and developing countries and the criteria used to classify hypertension”. What is the difference in term of ethnicity and race that explain this difference; i.e. what is the race/ethnicity in India and the race/ethnicity in the study by Moraes et al. In addition, also describe criteria to classify hypertension in both studies. And then discuss the mechanisms of these differences.

18) The comparison with the study by Noubiap and colleagues should be more detailed. The authors suggested Black Africans children in Africa had lower prevalence compared to Indian Adolescents. Authors may report prevalence data on hypertension in adults, and compare Indians and Black Africans to support this hypothesis (consider using global burden of diseases data) [https://jamanetwork.com/journals/jama/fullarticle/2596292]. Also add sentences on the mechanism of this difference: genetics? Dietary habits (with more details)?

Reviewer #2: General comment

Thanks for this review which brings new information. However, there are some concerns raised below. In addition, the paper should be thoroughly proofread: there are many grammatical errors and the style used makes sometimes the message hard to capture.

Specific comments

Lines 102-3: hard to understand what the authors are meaning

Line 105: This should be the beginning of a new paragraph, starting with “Adolescents…”

Line 109: The authors explain the rationale of their study by indicating that most studies that were carried out in India were school based, and that their review would help solving such inconsistencies. I totally disagree, as the review is just a summary of existing literature; limitations of individual studies constitute limitations of the review…

Lines 129-31: These sentences relate to study selection, not literature search strategy

Line 138: why only cross sectional studies were included? Baseline or end-of-study data of cohort studies could present prevalence estimates, as well as control arms of RCTs…

Line 138: The title is about a study in adolescents, but here the authors say they have included children and ados up to 19yo

Line 139: please what do you mean by enough data was available?

Line 140: why papers written in Hindi were not considered? All these exclusions may have tended to increase the probability of publication bias

Line 143-4: why were studies within which parents had specific conditions rather than their kids excluded?

The authors do not mention to have excluded studies like in pregnant ados, or those with other conditions capable of biasing the prevalence of HTN? Nevertheless, they could still have included those studies and present them n subgroup analysis

Lie 150: I do not align with eliminating duplicates without verifying which is the more recent and complete version before ruling out…

Line 176: I would have suggested to add age groups, setting (urban/rural), sex (male/female), and year of publication as other variables for subgroup analysis; social d; social development index/human development index or quintile of life are other parameters to consider, to know if poorer ados have more or less HTN than richer ones

Fig 1: there is a problem in this figure: records excluded should be linked with records after duplicates removed instead of records screened; in fact, it is after exclusion that the remaining records are screened

What do you mean by non-relevant articles, n=92? Does it mean that your first round of exclusion was too sensitive?

Table 1: Is putting the study area more informative than for eg the setting: urban vs rural? It is also important to know how HTN was defined in each of these studies; was the study multisite/national representative?

Line 209: majority is how many?

Line 217: how many studies were national representative, data were they collected prospectively or retrospectively?

Line 223; risk of bias should precede data on HTN

Table 2 can be sent as supplementary table or the final score can be included in table 1

Line 237: what guided the breakdown into 4 regions? This was not presented and clearly explained in methodology

Line 247: this is hard to understand: what was the region of reference, for the others to have significant different prevalence estimates?

Line 278: the reason given to justify why publication bias was not assessed in the whole sample of studies is not clear to me. Should this bias be assessed only when there is low heterogeneity?

Table 3: I would have appreciated to see more variables in the subgroup analysis; see comment above

Line 299 & table 5: how can males and females be included in the same model? This is to be the same variable, sex with two modalities male and female

Study setting, region, method of sampling, method of HTN definition, etc could be explored in the model

Line 323: no reference is given to support the statement on HTN and race; I am not sure the comparison between Noubiap and the authors is suitable, as Noubiap included children in their review

Line 328: what is the pertinence in this statement?

Lines 337-44: this should be also discussed as limitations of the review; refer to the above comment in the introduction

Lines 345-7: I have not seen this result earlier

Lines 411-12; these lines are very insufficient as limitations of this study; please see comments above; in addition, discuss the generalizability of finding to the entire country, heterogeneity, etc

Further, I think the discussion is insufficient: little was discussed about clinical implications of this study in terms of harmonizing the way of measuring BP and defining HBP among ados in the country; public health implications in terms of what strategies could be put in place to raise awareness on the issue and preventive measures/programs that could be introduced in school programs, etc

Some refs could be updated, eg ref 9 published in 2006

6. PLOS authors have the option to publish the peer review history of their article (what does this mean?). If published, this will include your full peer review and any attached files.

Reviewer #1: Yes: Jean Joel Bigna

Reviewer #2: No

---

## [Author Response · Author response to Decision Letter 0]

19 Apr 2020

I have updated the search till 1st March 2020.

Captions for supporting information is added now.

Reply to the reviewers’ comment

Prevalence of hypertension among adolescents (10-19 years) in India: A systematic review and meta-analysis of cross-sectional studies.

Reviewer Number Original comments of the reviewer Reply by the author(s) Changes done on (manuscript with track changes) page number and line number

1 I suggest to the authors to replace in the title the term “observational” by “cross-sectional” since in inclusion criteria, they considered only cross-sectional studies. We thank the reviewer for this correction. This is corrected now Page- 1, line-3

1 Methods: The authors should update the stage of the review in PROSPERO. For instance, “Preliminary searches” are not “completed” This is addressed now Page-7, line-138

1 Inclusion: It is not clear whether the authors only considered population/community-based studies (including school-based) studies. Only this kind of studies should be considered to have reliable epidemiological data This is addressed now Page-7, line-144.

1 Data extraction: Cochrane do not recommend scoring quality assessment We agree, sir. However, there are instances where Cochrane reviewers have used Newcastle Ottawa Scale in systematic reviews of observational studies. The following is one example of a Cochrane review where NOS has been used.

Reference: 

1. Lansbury LE, Rodrigo C, Leonardi-Bee J, Nguyen-Van-Tam J, Shen Lim W. Corticosteroids as Adjunctive Therapy in the Treatment of Influenza: An Updated Cochrane Systematic Review and Meta-analysis. Read Online: Critical Care Medicine | Society of Critical Care Medicine [Internet]. 2020 [cited 2020 Apr 17];48: e98. Page-8, line-168-175

1 Statistical analysis: Consider reporting all prevalence estimates with 95% prediction intervals that better measure uncertainty We believe confidence intervals are better suited for the present study, considering that the purpose of using prediction intervals is primarily for predicting future unobserved observation and for other reasons mentioned in the following citation:

Reference: 

Chiolero A, Santschi V, Burnand B, Platt RW, Paradis G. Meta-analyses: with confidence or prediction intervals? Eur J Epidemiol [Internet]. 2012 [cited 2020 Apr 17]; 27:823–5. Page-8

1 Results: Characteristics of included studies. The sentence “All included studies were cross-sectional studies.” is not necessary since in the Inclusion criteria, the authors stated that they will consider only cross-sectional studies This is addressed now Page-9, line-206

1 Table 1 should include diagnostic criteria for hypertension This is addressed now Page 11,12,13, Table 1.

1 Table 2: The signification of 0, 1, 2, 3, 4 for Selection, Comparability, and Outcome is not clear. This was not described any part in manuscript. This is addressed now Page-8, line-171-714

1 Please report all prevalence with only one decimal This is addressed now Entire document

1 “a little increase”. I’m not sure it is adequate to write a little increase. I would suggest changing to “no substantial change”. This is addressed now Page-21, line-320

1 “Analysis repeated after removing two studies that reported the hypertension prevalence of more than 15% and the prevalence of hypertension decreased from 7.64% (95% CI: 6.13% to 9.14) to 6.75% (95% CI: 5.55 to 7.96%) with more precise estimate.” It is not a correct methodological practice to exclude studies based on the outcome of outcome (here prevalence). Sensitivity analyses are carried out on methodological criteria, criteria for inclusion of participants and methodological quality (risk of bias). The authors may explore why these studies had higher prevalence when looking at participants characteristics, method (example sampling method), setting (urban vs rural for example)… Authors can conduct leave-one-out sensitivity analysis to assess the effect on the overall pooled estimate of removing each study. This is addressed now Page-21, line-321-328

1 “We also reran the main analysis by removing two community-based study and one hospital-based study”. It is not clear why the authors removed both community-based and hospital-based studies. This is addressed now Page-21, line-321-328

1 “The analysis was repeated after removing the study done by Maiti & Bandyopadhyay with sample size of less than 200 and the pooled estimate came out to be 7.57%, 95% CI: 6.04 to 9.09%.” The authors should give a strong rationale on the reason for choosing a threshold of 200. This is addressed now Page-21, line-321-328

1 Table 5. 

a. It is not whether the findings in this table are for univariable or multivariable meta-regression. Both models (uni- and multivariable) should be reported. b. I do not understand why some variance are reported with “(-)”. Variance is never negative. c. P value less than 0.05 should be reported with at least three decimals. d. Meta-regression should include methods to diagnose hypertension i.e. diagnostic criteria. e. For total no. of males and females, only the proportion of males or females should be considered since the no. can vary among studies, but the proportion is a comparable measure of the representativeness of sex distribution. f. How many studies were included in final multivariable model? g. How many studies were included in each univariable model? h. It is not clear why for the “Age group” variable, the authors reported only one coefficient since there would be n-1 coefficients according to the number age groups. All points raised for this table are addressed now Table 4, Page-22

1 Discussion: “The possible reason for disparity in the estimate of hypertension prevalence might be due to the race and ethnicity of the developed and developing countries and the criteria used to classify hypertension”. What is the difference in term of ethnicity and race that explain this difference; i.e. what is the race/ethnicity in India and the race/ethnicity in the study by Moraes et al. In addition, also describe criteria to classify hypertension in both studies. And then discuss the mechanisms of these differences. This is addressed now Page-23, line-355-361

1 The comparison with the study by Noubiap and colleagues should be more detailed. The authors suggested Black Africans children in Africa had lower prevalence compared to Indian Adolescents. Authors may report prevalence data on hypertension in adults and compare Indians and Black Africans to support this hypothesis (consider using global burden of diseases data) [https://jamanetwork.com/journals/jama/fullarticle/2596292]. Also add sentences on the mechanism of this difference: genetics? Dietary habits (with more details)? This is addressed now Page 23, line:365-372

2 hard to understand what the authors are meaning This is addressed now Page-5, line- 105.

2 This should be the beginning of a new paragraph, starting with “Adolescents…” This is addressed now Page-5, line-109

2 The authors explain the rationale of their study by indicating that most studies that were carried out in India were school based, and that their review would help solving such inconsistencies. I totally disagree, as the review is just a summary of existing literature; limitations of individual studies constitute limitations of the review… This is addressed now Page-6, line-113-116

2 These sentences relate to study selection, not literature search strategy This is addressed now Page-7, line:140-141.

2 why only cross-sectional studies were included? Baseline or end-of-study data of cohort studies could present prevalence estimates, as well as control arms of RCTs… We thank the reviewer for this comment. However, we believe that large cross-sectional studies would be the best design for this question, as even control arm of RCTs or cohort study population would not be representative. A cross-sectional study would be able to yield the best estimate for baseline risk.

Reference:

Kesmodel US. Cross-sectional studies – what are they good for? Acta Obstetricia et Gynecologica Scandinavica [Internet]. 2018 [cited 2020 Apr 7]; 97:388–93. Page-7, line:142

2 The title is about a study in adolescents, but here the authors say they have included children and ados up to 19yo This is addressed now Page-7, line:142-143

2 please what do you mean by enough data was available? This is addressed now Page-7, line:144-145

2 why papers written in Hindi were not considered? All these exclusions may have tended to increase the probability of publication bias There are no databases for Hindi studies and all the Indian papers are written in English. The following is the reference for all papers from India being published in English. For these reasons we believe that this will not be a source of publication bias.

Reference:

Medical Journals of India [Internet]. [cited 2020 Apr 17]. Available from: http://medind.nic.in/ Page-7, line:146

2 why were studies within which parents had specific conditions rather than their kids excluded? This is addressed now Page-7, line:120-152

2 I do not align with eliminating duplicates without verifying which is the more recent and complete version before ruling out This is addressed now Page-7, line:158-159

2 I would have suggested to add age groups, setting (urban/rural), sex (male/female), and year of publication as other variables for subgroup analysis; social d; social development index/human development index or quintile of life are other parameters to consider, to know if poorer ados have more or less HTN than richer ones This is addressed now Page-9, line:186-187

2 Figure1: there is a problem in this figure: records excluded should be linked with records after duplicates removed instead of records screened; in fact, it is after exclusion that the remaining records are screened What do you mean by non-relevant articles, n=92? Does it mean that your first round of exclusion was too sensitive? This is addressed now Figure 1

2 Table1: Is putting the study area more informative than for eg the setting: urban vs rural? It is also important to know how HTN was defined in each of these studies; was the study multisite/national representative? This is addressed now Table1, Page-11

2 majority is how many? This is addressed now Page-15, line:223

2 how many studies were national representative, data were they collected prospectively or retrospectively? This is addressed now Page-15,

line-230-231

2 Table 2 can be sent as supplementary table or the final score can be included in table 1 This is addressed now Table S2 (Supplement)

2 what guided the breakdown into 4 regions? This was not presented and clearly explained in methodology This is addressed now Page-9, line:186-187

2 this is hard to understand: what was the region of reference, for the others to have significant different prevalence estimates? This is addressed now Page-19, Line-286-287

2 the reason given to justify why publication bias was not assessed in the whole sample of studies is not clear to me. Should this bias be assessed only when there is low heterogeneity? This is addressed now Page-20, line:310-311

2 Table5: how can males and females be included in the same model? This is to be the same variable, sex with two modalities male and female Study setting, region, method of sampling, method of HTN definition, etc could be explored in the model This is addressed now Table 4, page-22

2 Discussion: no reference is given to support the statement on HTN and race; I am not sure the comparison between Noubiap and the authors is suitable, as Noubiap included children in their review This is addressed now Page-23, line-365-373

2 Discussion: what is the pertinence in this statement? This is addressed now Page-24, line:377-378

2 these lines are very insufficient as limitations of this study; please see comments above; in addition, discuss the generalizability of finding to the entire country, heterogeneity, etc This is addressed now Page-28, line:480-485

2 Further, I think the discussion is insufficient: little was discussed about clinical implications of this study in terms of harmonizing the way of measuring BP and defining HBP among ados in the country; public health implications in terms of what strategies could be put in place to raise awareness on the issue and preventive measures/programs that could be introduced in school programs, etc This is addressed now Page-27-28, line-458-472

---

## [Decision Letter · Decision Letter 1]

24 Jun 2020

PONE-D-20-01922R1

Prevalence of hypertension among adolescents (10-19 years) in India: A systematic review and meta-analysis of cross-sectional studies

PLOS ONE

Dear Dr. Haldar,

Thank you for submitting your manuscript to PLOS ONE. After careful consideration, we feel that it has merit but does not fully meet PLOS ONE’s publication criteria as it currently stands. Therefore, we invite you to submit a revised version of the manuscript that addresses the points raised during the review process.

Among the major issues you should address is the age range of participants in the studies included in the metaanalysis. For studies in which the age range falls beyond the range indicated in the study, you should either extract data of eligible participants or exclude the study if extraction is not possible.  

We look forward to receiving your revised manuscript.

Kind regards,

Simeon-Pierre Choukem

Academic Editor

PLOS ONE

Reviewers' comments:

Reviewer's Responses to Questions

**Comments to the Author**

1. If the authors have adequately addressed your comments raised in a previous round of review and you feel that this manuscript is now acceptable for publication, you may indicate that here to bypass the “Comments to the Author” section, enter your conflict of interest statement in the “Confidential to Editor” section, and submit your "Accept" recommendation.

Reviewer #1: (No Response)

Reviewer #3: (No Response)

2. Is the manuscript technically sound, and do the data support the conclusions?

Reviewer #1: No

Reviewer #3: Yes

3. Has the statistical analysis been performed appropriately and rigorously? 

Reviewer #1: Yes

Reviewer #3: No

4. Have the authors made all data underlying the findings in their manuscript fully available?

Reviewer #1: Yes

Reviewer #3: Yes

5. Is the manuscript presented in an intelligible fashion and written in standard English?

Reviewer #1: Yes

Reviewer #3: Yes

6. Review Comments to the Author

Reviewer #1: I thank the authors for this important work. I appreciated their responses to queries. However, this review and meta-analysis should be extensively revised. The authors should perform what they stated they will perform in Methods section. There are two major concerns and this study should not be considered for publication until strict respect to the methodology.

1) The authors stated that they will included studies conducted among adolescents, 10 to 19 years. However, among studies included, some of them included participants < 10 years old: Anjana 2005 (6-14 years), Buch 2011 (6-18 years), Mujumdar 2012 (6-15 years), Yuvaraj 2014 (9-16 years), George 2014 (9-18 years), Kumar 2017 (5-14 years), and Rai 2018 (8-17 years). Definitively, this is not a study a meta-analysis of data from adolescents since the authors included data from children/infants. The authors should exclude these studies from the review or when possible extract data from adolescents only (> or = 10 years) (if not possible, then exclude).

2) The authors stated that they will include only population/community-based (including school-based) studies. This is an important criteria and I agree with this. However, the authors included a hospital-based study: Soundarssanane 2006.

Reviewer #3: 1. This manuscript describes a systematic review and meta-analysis which aims to calculate the pooled prevalence of hypertension among adolescents (10-19 years) in India. The research is valuable given the well-known short-term and long-term ill effects of high blood pressure diagnosed in adolescence. I believe this study could be a useful contribution to the literature examining the global burden of hypertension in adolescents.

2. The main title clearly identifies the report as a systematic review and meta-analysis; it also indicates the target population as well as the epidemiological measure (“prevalence”) that will be pooled. However, the running title does not contain the world “prevalence”. We suggest that “prevalence” should be added to the running title to fully match with the main title.

3. The abstract summarizes the rationale for the study and the analytic methods used, as well as the key findings.

#Background (line 43): The authors are right that there are concerns on the threat of elevated blood pressure in children and adolescents. It is also known that these concerns originate from existing evidence on the short-term and long-term ill effects of high blood pressure diagnosed in adolescence. This suggests that the authors could replace the expression “…considerable concerns on the threat…” by something like “… the well-known short-term and long-term ill effects…”.

#Methods: The processes of study search and selection were described. However, there is no mention that the study only included manuscripts written in English language (lines 48-50); it is a detail worth mentioning. Further, it is not stated that a meta-regression analysis was carried out, whereas the authors took great care to do it; this should be added to the abstract to accurately reflect the analyses performed during the research.

#Results: The abstract contains the value of the pooled prevalence. It would have been interesting if it also contained the range of the prevalence across studies, so that readers could have an idea of the variability of such measure. Besides, the value of the heterogeneity of subgroup analyses was described as “acceptable” (line 59). It is understandable that a low level of heterogeneity can be deemed acceptable. However, it might be preferable to use the established thresholds of heterogeneity which will qualify an I-square = 18.3% as “small”.

#Conclusion: The authors proposed legitimate policy measures that should be put in place to control hypertension in adolescents. These recommendations could have been stronger if they were preceded by a comment describing the value of that pooled prevalence as “high”.

#Key words (line 65): The abstract contains the key words that will facilitate future bibliographical searches. “Systematic review” and “prevalence” are other important key words that were omitted and ought to be added.

4. The introduction includes important information about the global burden of elevated blood pressure, the rising trend of hypertension in the adolescent population contrasted by the paucity of evidence on its burden and the low level of awareness, the considerable proportion of India’s population that is adolescent, and the lack of information on the national prevalence of hypertension in that population.

# (lines 88-90) The authors supported their point by raising awareness about the burden of cardiovascular diseases (CVD) in developing countries. The expression “…which is now being recognized as a major killer in developing countries” conveys the same information as the first part of the sentence and thus should be removed. Furthermore, the argument about high BP in adolescents of developing countries could have been strengthened by providing factual evidence such as the value of 5.5% that represents the pooled prevalence of elevated blood pressure in children and adolescents in Africa computed by Noubiap JJ et al.

# (lines 100-104) An attempt was made to provide evidence of the link between blood pressure in adolescence and hypertension in adulthood, which is commendable. We believe that more detailed information on this link should be presented to further support the rationale of this study.

# (lines 104-106) Childhood blood pressure (BP) was presented as the ”best” indicator of adult BP. The quality of the reference used is not contested. However, the use of the word “best” is questionable, because it supposes that in this reference, measures of association were used to quantify the predictive value of childhood BP; it also suggests that a comparison was made with other potential predictive factors. In their report, Xiaoli Chen and Youfa Wang did a meta-regression analysis of correlation coefficients rather than measures of association. Further, they did not make a comparison with the predictive value of other potential predictive factors such as diet, family history, etc… Therefore, it would be more accurate to rather describe childhood BP as a “strong” indicator of adult BP.

# (lines 116-123) The importance of the current research is justified by arguing that there is a lack of studies reporting on the prevalence of hypertension among adolescents; it is also claimed that there exists no national survey investigating that issue. It is important to remind that the current work is not an original research but rather a review of existing studies; furthermore, it is not a national survey. Therefore, alternate arguments should be found to justify the relevance of the present study.

5. The methods provides details about the literature search strategy, the study selection criteria and selection process, the data extraction, and the statistical analyses.

# (lines 163-167) The process used to extract data was presented clearly. However, it is not possible to know if there were attempts made to contact the authors of publications that had missing information on key variables. This is critical given that 3,326 participants had missing information on sex (line 208), and 3 studies did not mention the number of BP readings (line 291).

# (lines 168-174) The quality of each included study was assessed using an established scale. The 2 sentences spanning from line 168 to line 174 convey the same information and hence should be combined.

# (lines 182-183) There was an attempt to measure the heterogeneity between the included studies. The only metric used was the I-square. Although we do not contest that the I-square must be presented, using it solely is not convenient for several reasons:

(1) the I-square is not a test statistic but rather a scale;

(2) the I-square does not measure the heterogeneity itself but rather provides the percentage of variation

across studies that is due to their heterogeneity rather than chance;

(3) there is much uncertainty about the I-square when there are few studies such as is the case here

(https://training.cochrane.org/handbook/current/chapter-10#section-10-10-2);

(4) the established test statistic to confirm the presence of heterogeneity is the Cochrane's Q statistic.

Therefore, in addition to the I-square, a Cochrane's Q statistic test of heterogeneity should also be performed, and the results presented and discussed.

# (line 183) The result of the pooled analysis is referred to as the “mean” percent prevalence. It is agreed that the pooled analysis uses a sort of average across studies, but it is in no case a simple mean since there are specific weights that are applied. Therefore, the term “mean” should be replaced by “pooled”.

# (lines 198-192) A meta-regression analysis was carried out to identify the causes of heterogeneity. Two important information were not provided:

(1) how the model fit was assessed;

(2) how the researchers controlled for false-positive findings (type I error) when performing meta-

regression with multiple covariates.

This information should be provided to reinforce the robustness of the analyses seen throughout the paper.

6. The data appear to be sound and the results section presents the output of the processes described in the methods section.

# (line 212) Table 1 presents the characteristics of the studies included in the systematic review and meta-analysis. It appears that several of these studies included participants with an age that fell outside the target range of 10-19 years. This raises several questions about the following:

(1) the accuracy of the study selection process, especially whether the inclusion criteria were respected;

(2) the rationale of restricting the meta-analysis to the adolescent population given the rarity of studies

treating about hypertension in the youth of India; this further raises concerns about the main title of the

systematic review.

# (line 206-207) It should be written: “The total number …was 28,355…”

# (line 250) As stated above, the term “mean” should be replaced by “pooled”.

# (line 252) As stated above, the I-square is not a heterogeneity test but rather a simple scale. The Cochrane's Q statistic test should be performed, and the results presented and discussed.

7. The discussion and conclusion are well balanced and adequately supported by the data.

# (Paragraph starting on line 352) There are valuable arguments presented to explain why the value of the pooled estimate is different from what other authors had found. It would have been easier for the readers to follow the trend of thoughts if there was first a formal comparison made between these values, before presenting the reasons for any difference observed.

# (line 392-393) Community-based studies were described as “superior” to school-based studies. The manuscript lacks the criteria on which this superiority was assessed as well as the references.

# (line 380-381) Specific references should be provided for the criteria of the European Hypertension society and the American Academy of Pediatrics, respectively.

# (line 411-428) The authors did well by highlighting that some of the differences observed might be due to the difference in blood pressure measurement methods used in various studies. However, a more thorough discussion on these methods is warranted because of the great influence that they might have on the accuracy of the diagnosis of hypertension.

7. PLOS authors have the option to publish the peer review history of their article (what does this mean?). If published, this will include your full peer review and any attached files.

Reviewer #1: **Yes: **Jean Joel Bigna

Reviewer #3: No

---

## [Author Response · Author response to Decision Letter 1]

10 Aug 2020

Dear Sir, 

We have responded to all the comments given by the reviewers and the manuscript has been revised. 

Response to reviewers

Prevalence of hypertension among adolescents (10-19 years) in India: A systematic review and meta-analysis of cross-sectional studies.

Reviewer number Original comments of the reviewer Reply by the Author(s) Changes done on page number and line number

1 The authors stated that they will be included studies conducted among adolescents, 10 to 19 years. However, among studies included, some of them included participants < 10 years old: Anjana 2005 (6-14 years), Buch 2011 (6-18 years), Mujumdar 2012 (6-15 years), Yuvaraj 2014 (9-16 years), George 2014 (9-18 years), Kumar 2017 (5-14 years), and Rai 2018 (8-17 years). Definitively, this is not a study a meta-analysis of data from adolescents since the authors included data from children/infants. The authors should exclude these studies from the review or when possible extract data from adolescents only (> or = 10 years) (if not possible, then exclude). Thank you for the comments. Yes, indeed as per inclusion criteria, we did include only the studies where it was possible to abstract data for ages 10 to 19 years. Page 7, line 126-128.

1 The authors stated that they will include only population/community-based (including school-based) studies. This is an important criterion and I agree with this. However, the authors included a hospital-based study: Soundarssanane 2006 Thank you so much. We have addressed this issue now. One hospital-based study is removed.

Accordingly, we updated our meta-analysis. Page 10, updated table 1. After excluding study author Soundarssanane et al

3 The main title clearly identifies the report as a systematic review and meta-analysis; it also indicates the target population as well as the epidemiological measure (“prevalence”) that will be pooled. However, the running title does not contain the world “prevalence”. We suggest that “prevalence” should be added to the running title to fully match with the main title. Thank you so much. This was an oversight from us. We have added the word ‘prevalence in title’ Page 1, line 4.

3 Abstract- Background (line 43): The authors are right that there are concerns on the threat of elevated blood pressure in children and adolescents. It is also known that these concerns originate from existing evidence on the short-term and long-term ill effects of high blood pressure diagnosed in adolescence. This suggests that the authors could replace the expression “…considerable concerns on the threat…” by something like “… the well-known short-term and long-term ill effects…”. Thank you for this suggestion. We have done correction accordingly in the abstract. Page 3, line 39.

3 Methods: The processes of study search and selection were described. However, there is no mention that the study only included manuscripts written in English language (lines 48-50); it is a detail worth mentioning. Further, it is not stated that a meta-regression analysis was carried out, whereas the authors took great care to do it; this should be added to the abstract to accurately reflect the analyses performed during the research. Thank you for this comment. This is addressed now Page 3, line 46 & 51.

3 Results: The abstract contains the value of the pooled prevalence. It would have been interesting if it also contained the range of the prevalence across studies, so that readers could have an idea of the variability of such measure. Besides, the value of the heterogeneity of subgroup analyses was described as “acceptable” (line 59). It is understandable that a low level of heterogeneity can be deemed acceptable. However, it might be preferable to use the established thresholds of heterogeneity which will qualify an I-square = 18.3% as “small”. Thank you for this suggestion. We have done correction accordingly in the abstract. Page 3, line 54-57.

3 Conclusion: The authors proposed legitimate policy measures that should be put in place to control hypertension in adolescents. These recommendations could have been stronger if they were preceded by a comment describing the value of that pooled prevalence as “high”. This is addressed now. We have now removed this recommendation from the manuscript. Page 4, line 60-61.

3 #Key words (line 65): The abstract contains the key words that will facilitate future bibliographical searches. “Systematic review” and “prevalence” are other important key words that were omitted and ought to be added. Thank you for this suggestion. We have done correction accordingly in the keywords. Page 4, line 62.

3 (lines 88-90) The authors supported their point by raising awareness about the burden of cardiovascular diseases (CVD) in developing countries. The expression “…which is now being recognized as a major killer in developing countries” conveys the same information as the first part of the sentence and thus should be removed. Furthermore, the argument about high BP in adolescents of developing countries could have been strengthened by providing factual evidence such as the value of 5.5% that represents the pooled prevalence of elevated blood pressure in children and adolescents in Africa computed by Noubiap JJ et al. Thank you for this comment. This is corrected now in the introduction. Page 5, line 79 and line 86-87.

3 (lines 100-104) An attempt was made to provide evidence of the link between blood pressure in adolescence and hypertension in adulthood, which is commendable. We believe that more detailed information on this link should be presented to further support the rationale of this study. Thank you so much for this comment. This is addressed now. Page 5, line 94-97.

3 (lines 104-106) Childhood blood pressure (BP) was presented as the” best” indicator of adult BP. The quality of the reference used is not contested. However, the use of the word “best” is questionable, because it supposes that in this reference, measures of association were used to quantify the predictive value of childhood BP; it also suggests that a comparison was made with other potential predictive factors. In their report, Xiaoli Chen and Youfa Wang did a meta-regression analysis of correlation coefficients rather than measures of association. Further, they did not make a comparison with the predictive value of other potential predictive factors such as diet, family history, etc… Therefore, it would be more accurate to rather describe childhood BP as a “strong” indicator of adult BP. Thank you for this comment. This is addressed now. Page 5, line 98

3 (lines 116-123) The importance of the current research is justified by arguing that there is a lack of studies reporting on the prevalence of hypertension among adolescents; it is also claimed that there exists no national survey investigating that issue. It is important to remind that the current work is not an original research but rather a review of existing studies; furthermore, it is not a national survey. Therefore, alternate arguments should be found to justify the relevance of the present study Thank you so much for this comment. This is addressed now in the rationale part. Page 6, line 105.

3 (lines 163-167) The process used to extract data was presented clearly. However, it is not possible to know if there were attempts made to contact the authors of publications that had missing information on key variables. This is critical given that 3,326 participants had missing information on sex (line 208), and 3 studies did not mention the number of BP readings (line 291). Authors were not contacted. 

 -

3 (lines 168-174) The quality of each included study was assessed using an established scale. The 2 sentences spanning from line 168 to line 174 convey the same information and hence should be combined. Thank you for this comment. This is addressed now. Page 8, line 150-156.

3 (lines 182-183) There was an attempt to measure the heterogeneity between the included studies. The only metric used was the I-square. Although we do not contest that the I-square must be presented, using it solely is not convenient for several reasons: (1) the I-square is not a test statistic but rather a scale; (2) the I-square does not measure the heterogeneity itself but rather provides the percentage of variation across studies that is due to their heterogeneity rather than chance; (3) there is much uncertainty about the I-square when there are few studies such as is the case here (https://training.cochrane.org/handbook/current/chapter-10#section-10-10-2); (4) the established test statistic to confirm the presence of heterogeneity is the Cochrane's Q statistic. Therefore, in addition to the I-square, a Cochrane's Q statistic test of heterogeneity should also be performed, and the results presented and discussed. Thank you so much for this suggestion. Cochrane's Q statistic test of heterogeneity is reported throughout the paper now. Page 8, line 164

3 (line 183) The result of the pooled analysis is referred to as the “mean” percent prevalence. It is agreed that the pooled analysis uses a sort of average across studies, but it is in no case a simple mean since there are specific weights that are applied. Therefore, the term “mean” should be replaced by “pooled”. Thank you for the comment. This is addressed throughout the paper now. Page 8, line 166

3 (lines 198-192) A meta-regression analysis was carried out to identify the causes of heterogeneity. Two important information were not provided: (1) how the model fit was assessed; (2) how the researchers controlled for false-positive findings (type I error) when performing meta- regression with multiple covariates. This information should be provided to reinforce the robustness of the analyses seen throughout the paper. Thank you so much for this valuable suggestion. This is addressed now.

1.) R2 value is reported for assessment of goodness of fit.

2.) Monte Carlo permutation test was conducted and the adjustment for type 1 error did not changed the result. Page 9, line 177-179.

3 (line 212) Table 1 presents the characteristics of the studies included in the systematic review and meta-analysis. It appears that several of these studies included participants with an age that fell outside the target range of 10-19 years. This raises several questions about the following: (1) the accuracy of the study selection process, especially whether the inclusion criteria were respected; (2) the rationale of restricting the meta-analysis to the adolescent population given the rarity of studies treating about hypertension in the youth of India; this further raises concerns about the main title of the systematic review. Thank you for the comments. Yes, 

1.) As we have stated before we have included only those studies where it was possible to abstract data for ages 10 to 19 years. Thus, this suggestion is already incorporated in our review.

2.) We intended for results to be applicable to the adolescent age group since it represents a distinct demographic and a unique stage of development. For this reason, we included studies from which we could abstract data specific to this age group, and the title reflects this. -

3 (line 206-207) It should be written: “The total number …was 28,355…” Thank you for this comment. This is addressed now Page 9, line 189

3 (line 250) As stated above, the term “mean” should be replaced by “pooled”. Thank you for this comment. This is addressed now Page 15, line 216

3 (line 252) As stated above, the I-square is not a heterogeneity test but rather a simple scale. The Cochrane's Q statistic test should be performed, and the results presented and discussed. Thank you for this comment. As mentioned above we have addressed this throughout the manuscript. Page 15 and page 17 – Table 2 and 3.

3 (Paragraph starting on line 352) There are valuable arguments presented to explain why the value of the pooled estimate is different from what other authors had found. It would have been easier for the readers to follow the trend of thoughts if there was first a formal comparison made between these values, before presenting the reasons for any difference observed. Thank you for this comment. This is addressed now. Page 19, line 282.

3 (line 392-393) Community-based studies were described as “superior” to school-based studies. The manuscript lacks the criteria on which this superiority was assessed as well as the references. Thank you so much for this comment. This is addressed now. Page 20, line 316-318.

3 (line 380-381) Specific references should be provided for the criteria of the European Hypertension society and the American Academy of Pediatrics, respectively. Thank you for this comment. The references have been added now. Page 20, line 307.

3 (line 411-428) The authors did well by highlighting that some of the differences observed might be due to the difference in blood pressure measurement methods used in various studies. However, a more thorough discussion on these methods is warranted because of the great influence that they might have on the accuracy of the diagnosis of hypertension. Thank you so much for suggestion. This is addressed now. Page 21, Line 347-367.

---

## [Decision Letter · Decision Letter 2]

16 Sep 2020

Prevalence of hypertension among adolescents (10-19 years) in India: A systematic review and meta-analysis of cross-sectional studies.

PONE-D-20-01922R2

Dear Dr. Haldar,

We’re pleased to inform you that your manuscript has been judged scientifically suitable for publication and will be formally accepted for publication once it meets all outstanding technical requirements.

Kind regards,

Simeon-Pierre Choukem

Academic Editor

PLOS ONE

Additional Editor Comments (optional):

Reviewers' comments:

Reviewer's Responses to Questions

**Comments to the Author**

1. If the authors have adequately addressed your comments raised in a previous round of review and you feel that this manuscript is now acceptable for publication, you may indicate that here to bypass the “Comments to the Author” section, enter your conflict of interest statement in the “Confidential to Editor” section, and submit your "Accept" recommendation.

Reviewer #1: All comments have been addressed

Reviewer #3: All comments have been addressed

2. Is the manuscript technically sound, and do the data support the conclusions?

Reviewer #1: Yes

Reviewer #3: Yes

3. Has the statistical analysis been performed appropriately and rigorously? 

Reviewer #1: Yes

Reviewer #3: Yes

4. Have the authors made all data underlying the findings in their manuscript fully available?

Reviewer #1: Yes

Reviewer #3: Yes

5. Is the manuscript presented in an intelligible fashion and written in standard English?

Reviewer #1: Yes

Reviewer #3: Yes

6. Review Comments to the Author

Reviewer #1: I thank you for the revision of the paper. All my queries have been adressed. The paper is now acceptable for publication.

Reviewer #3: General: We commend the authors for the extensive work that was done to fully address our concerns.

Line 274: We suggest that the authors replace the word "countries" by "regions".

Lines 275-277: The authors indicate the cut off percentile (90th) used as a reason of the difference between that results obtained by Moraes and theirs. We suggest that the authors also indicate the cut off that was used in their study so that the readers will be able to do a direct comparison.

Lines 401-410: The authors might also want to add to their limitations the fact that it is a very small number of studies conducted in rural settings that was included in their study, as this is an important determinant of the generalizability of their results.

Lines 59-60 and 412-413: We believe that the authors should maintain the recommendations that were in the previous version, for two reasons: (1) our comments did not judge the said recommendations as invalid, but rather proposed that the recommendations be backed by a description of the results; (2) the said recommendations are warranted given the fact that the pooled estimate obtained might actually be an underestimate of the true figure.

7. PLOS authors have the option to publish the peer review history of their article (what does this mean?). If published, this will include your full peer review and any attached files.

Reviewer #1: **Yes: **Jean Joel Bigna

Reviewer #3: No

---

## [Editor Report · Acceptance letter]

24 Sep 2020

PONE-D-20-01922R2 

Prevalence of hypertension among adolescents (10-19 years) in India: A systematic review and meta-analysis of cross-sectional studies. 

Dear Dr. Haldar:

I'm pleased to inform you that your manuscript has been deemed suitable for publication in PLOS ONE. Congratulations! Your manuscript is now with our production department. 

Kind regards, 

on behalf of

Dr. Simeon-Pierre Choukem 

Academic Editor

PLOS ONE